# Regio- and enantioselective synthesis of acyclic quaternary carbons via organocatalytic addition of organoborates to (Z)-Enediketones

Po-Kai Peng [1], Andrew Isho[1] & Jeremy A. May [1] ✉

The chemical synthesis of molecules with closely packed atoms having their bond coordination saturated is a challenge to synthetic chemists, especially when three-dimensional control is required. The organocatalyzed asymmetric synthesis of acyclic alkenylated, alkynylated and heteroarylated quaternary carbon stereocenters via 1,4-conjugate addition is here catalyzed by 3,3´-bis-perfluorotoluyl-BINOL. The highly useful products (31 examples) are produced in up to 99% yield and 97:3 er using enediketone substrates and potassium trifluoroorganoborate nucleophiles. In addition, mechanistic experiments show that the (Z)−isomer is the reactive form, ketone rotation at the site of bond formation is needed for enantioselectivity, and quaternary carbon formation is favored over tertiary. Density functional theory-based calculations show that reactivity and selectivity depend on a key n→π* donation by the unbound ketone's oxygen lone pair to the boronate-coordinated ketone in a 5-exo-trig cyclic ouroboros transition state. Transformations of the conjugate addition products to key quaternary carbon-bearing synthetic building blocks proceed in good yield.

All-carbon quaternary stereocenters are an important synthetic motif found in natural products and bioactive molecules (Fig. 1) that are especially difficult to synthesize enantioselectively[1–4]. Successful strategies have recently been developed for cyclic systems (Fig. 2A)[2,5–10]; however, constructing quaternary centers in acyclic molecules remains a significant synthetic challenge due to the combination of high levels of steric congestion and greater conformational freedom[11–21]. Despite its known benefits, organocatalysis[22–24] has been used for asymmetric quaternary center construction only a few classes of acyclic systems[25–28], though asymmetric 1,4-conjugate addition for tertiary carbon synthesis is well documented[29–33]. This report details the successful organocatalytic synthesis of valuable acyclic 1,4-dicarbonyl products with vinylated and arylated quaternary centers[34].

Organocatalyzed Michael additions to acyclic proquaternary substrates have been reported for nitromethane or cyanide but have

otherwise been rare (Fig. 2B)[35–42]. A newer class of easily synthesized BINOL−derived enantioselective 1,4−addition organocatalysts have proven to be useful, recyclable, and functional group tolerant in many transformations[43–53]; however, these reactions have only produced chiral tertiary carbon centers to date. In fact, β,β−disubstituted enones were investigated for quaternary carbon formation but completely lacked reactivity[54].

## Results and discussion

To overcome steric deactivation via increased electrophilic activation, we looked to enones bearing additional electron-withdrawing groups. In particular, the use of 2−ene−1,4−diones (1) could allow an approach to often difficult-to-access chiral 1,4-diketones with beta quaternary carbons (Fig. 2C) and functionality for the total synthesis of natural products[55–57]. However, few precedents related to such a

[1]Department of Chemistry, University of Houston, 3585 Cullen Blvd., Fleming Building Rm 112, Houston, TX 77204-5003, USA. ✉e-mail: jmay@uh.edu

**Fig. 1 | Representative natural products bearing alkenylated quaternary carbons.** Inset shows how quaternary carbon formation applies to a synthesis[86].

**A) Cyclic quaternary carbon formation**

**B) Organocatalytic Michael addition**

**C) This work:**

**Fig. 2 | Enantioselective synthesis of quaternary carbons. A** Organometallic catalysis for cyclic quaternary carbon synthesis. **B** Organocatalytic Michael addition of nitromethane to chalcones. **C** The synthesis of acyclic quaternary carbons with alkenyl, alkynyl, and aryl substituents (orange "R´" substituent) via an ouroboros transition state discussed herein.

β-vinylation or arylation of ketones to construct quaternary centers exist[58–63].

To test the diketone activation hypothesis, enediketone **1a** was synthesized as a mixture of cis and trans isomers, which was then purified, and potassium styrenyl trifluoroborate was chosen as an exploratory nucleophile (Fig. 3). Unsurprisingly, the *E*−isomer was almost completely unreactive (<2% yield, ~77:23 er), so further experimentation was conducted with pure (*Z*)−enediketone. While we fully expected competing regioselectivity with addition at both alkene positions, substrate (*Z*)−**1a** reacted with superb regioselectivity with the use of typical conditions for BINOL 4−catalyzed conjugate additions[9,10] to give a single product. In fact, (*Z*)−**1a** defied all initial expectations of addition at the less hindered carbon and provided quaternary carbon-bearing 1,4-diketone **3a** in 99% yield and 96:4 er. A control experiment without any catalyst gave **3a** from (*Z*)−**1a** in 69% yield after 18 h, indicating that a significant racemic background reaction was operative. These results suggested that two cis-disposed ketone carbonyls must be present for reactivity and may provide *Z*-dependent cooperative activation. We were then able to demonstrate a one-pot reaction using (*E*)−**1a**, which was first converted to the *Z*-form

via photo-isomerization[64] and then could undergo conjugate addition to provide the quaternary carbon product in good yield (79%) and the same er obtained from pure (*Z*)-enediketone. Such an isomerization may even have been the source of activity seen in Fig. 3A. The presence or absence of light had no effect on the reaction using pure *Z*-enediketone. Various BINOL derivatives were tested for increased reactivity and stereoselectivity, and the effects of solvent and temperature were also investigated, but with no improvement (See "Synthesis of BINOL-based catalysts" in the Supplementary Information).

Various enediketones showed productive reactivity with the catalytic conditions identified above. Both electron−donating and electron−withdrawing groups on the ketone's aromatic ring provided effective reaction (**3a**–**3e**, Fig. 4). A heteroaromatic enone substituent was likewise accommodated (see **3f**). However, changing the phenylketone to a methylketone resulted in the formation of product **3g** with slightly reduced stereoselectivity and competing regioselectivity to form a tertiary stereocenter in 11% yield (see **5g**). Despite both carbonyls being equally Lewis basic, quaternary carbon formation was still favored in a 4:1 ratio. Interestingly, moving the branching vinyl methyl from the alkyl ketone side of the alkene to the phenyl ketone side reversed the regioselectivity so that the major diastereomer of dione **5h** was formed with low enantioselectivity in 63% yield. The minor diastereomer of **5h** was produced along with the quaternary product in 17% yield as an inseparable 2:1 mixture. An initial regioselectivity hypothesis was that the relative locations of the phenyl and aliphatic ketones have a directing effect on the addition, where C–C bond formation is more strongly favored at the β-carbon of an aryl ketone than that of an alkyl ketone[43–53]. Based on work by Goodman and Pellegrinet[44], we believed that the phenyl ketone directed an intramolecular 5-exo-trig 1,4-addition that reinforced the favored quaternary carbon formation.

Replacing the methyl groups of **1a** with ethyl groups gave improved stereoselectivity (**3i**) but slightly lowered the yield, likely due to an increase in steric repulsion. Changing the methyl ketone of **1a** to a phenyl ketone afforded the quaternary center in **3j** with moderate er; however, replacing both methyl groups with phenyls precluded reaction so triphenyl product **3k** was not observed. The series **3a, 3j**, and **3k** shows the negative impact of increasing the size of substituents on yield. It is noteworthy that a cyclic diketone system gave the alpha quaternary center in **3l** in high yield but with little enantioselectivity. Apparently, without rotation of the carbonyl-olefin bond where C–C bond formation occurs reactivity is retained, but the stereoselectivity is almost completely lost.

Various vinyl, alkynyl, and heteroaromatic nucleophiles were also examined. A few substrates gave a lower er, but use of trifluorotoluene and/or an increased loading of the catalyst improved the enantioselectivity (Fig. 5). For example, the electron rich styrenyl nucleophiles

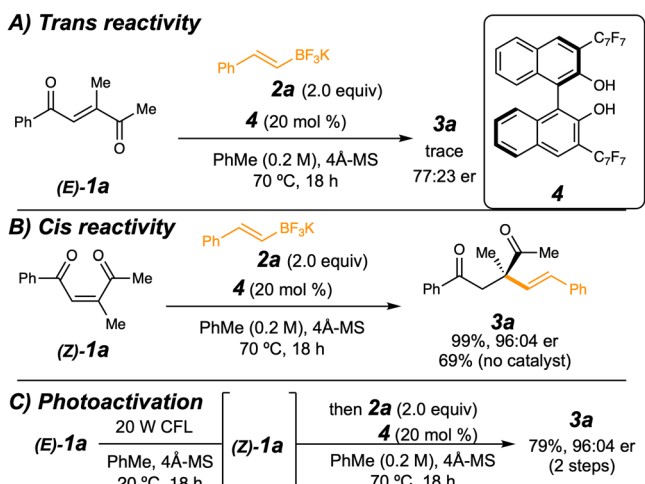

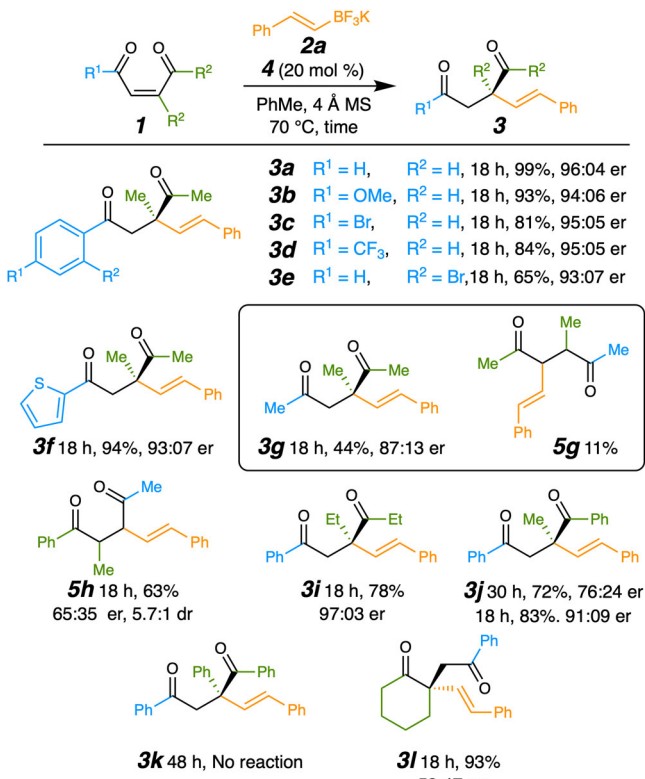

**Fig. 3 | Preliminary results and control experiments. A** Nearly no reactivity was observed with the *trans* diketone. **B** The *cis* substrate afforded catalytic activity that produced the acyclic quaternary carbon product in high yield and with high stereoselectivity despite a significant background reaction also being operative. **C** Pure *trans* diketone or a mixture of *cis* and *trans* could be converted to *cis*-enriched substrate that then reacted in a similar manner to pure *cis* diketone. 4Å-MS 4Å molecular sieves, PhMe toluene, er enantiomeric ratio, CFL Compact Fluorescent Lightbulb.

that afforded **3m** and **3n** originally showed a significant racemic background reaction, but increasing the catalyst loading improved the er to 80:20 and 92:8, respectively. An electron-withdrawing group on the styrene system in **3o** produced a high yield and er without adjustment. Nucleophiles with alkyl chains gave the dienyl adducts **3p, 3q**, and trans alkenyl **3r–3x** in high yield and enantioselectivity. Having two vinyl substituents resulted in diminished enantioselectivity (see **3y** and **3z**), but a synthetically useful bromo vinyl borate synthesized **3aa** in moderate yield and improved er. Alternatives to the vinyl nucleophilic system were also tested. Alkynyl reagents provided useful reactivity, but decreased stereoselectivity (see **3ab** and **3ac**). It is worth noting that the isomerization from (*Z*)−**1a** to (*E*)−**1a** occurred competitively during the formation of **3z** and **3ab**, which may have reduced both the yield and stereoselectivity for those reactions. The use of other strong nucleophiles, like furanyl borate, similarly formed quaternary carbons in high yield but with low enantioselectivity due to the competitiveness of the background reaction (see **3ad**)[65]. On the other hand, a thienyl borate produced **3ae** in good yield and high er.

To explain (A) why only the *Z*-isomer was reactive and (B) why quaternary regioselectivity was favored over tertiary carbon formation, we pursued a computational investigation using substrates (*Z*)−**1a** and (*E*)−**1a** with styrenyl boronate. Based on prior mechanistic investigations relevant to tertiary carbon formation via similar catalysis[47], it is likely that the potassium trifluoroborate salt dissociates fluoride and condenses with the BINOL **4** to form an activated chiral boronate ester that then coordinates to the enone carbonyl. To simplify the calculations, BINOL **4** was modelled as 3,3′-difluorobisphenol. We initially modeled the formation of the Lewis acid/Lewis base complexed boronate-ketone adduct **6**, and our calculations supported Goodman's finding[44] that this complex formed as a discrete mechanistic intermediate prior to the transition state (Fig. 6). Note that this stable intermediate was taken as the zero-point reference for all other calculated geometries. Conjugate addition transition states derived from ketone-coordinated boronates with both endo and exo modes of addition[66,67] were next examined (Figs. 6 and 7). Where Goodman's work showed 6-endo cyclization (see **8c** and **8e**), we found that 5-exo modes[68] were lower in energy for both quaternary and tertiary carbon formation (compare **8a** to **8c** and **8d** to **8e**). This new mode of

**Fig. 4 | Scope of products from various enediketones.** Reaction yields are of purified isolated products. Enantiomeric ratios were determined by HPLC with chiral stationary phase. For **5g** and **5h** the stereochemistry was not determined. For **3j** the use of both 20 mol % (30 h) and 40 mol % (18 h) of **4** is illustrated.

reactivity is enabled by the additional ketone. Close examination of the exo transition states revealed a fascinating stabilizing effect; the ketone distal to the Lewis acid coordination not only enabled the 5-exo addition but also participated in an n→π* donation to the bound carbonyl (**8a**)[69]. This ouroboros-like activation[70,71] is evidenced by the short C = O→C = O bond (1.53 Å in **8a**) and the tetrahedral geometry of the carbon of the bound C = O (C16). Such interactions have been described for static protein structure[72] and utilized for the synthesis of Lewis acid/base heteroaromatics[69], but to our knowledge has not been proposed as a stabilizing factor in reaction catalysis[73]. The LUMO of the C16 carbonyl thus acts as a Lewis acid activating the planar enone for 5-exo-trig conjugate addition, lowering the LUMO energy, and the electron donation of the planar ketone to the C16 carbonyl simultaneously increases the electron density in the nucleophilic system (C28), raising the HOMO energy. Additionally, this stabilizing interaction was not accessible in the 6-endo geometries (see longer O to C distances in **8c** and **8e**), and we believe this to be a reason for their relatively higher energy pathways. In investigating the generation of the ouroboros stabilization, we could identify that the formation of iso-furan **7** occurred prior to C−C bond formation. Some pathways, such as that shown in Fig. 6, have **7** formed as a meta stable intermediate as a local minimum. In others, it is a shoulder or part of a continuous slope to the transition state. These calculations also showed that the (*R*)-biaryl introduces torsion in the coordinated system that favors **8a** over **8b**, giving the major observed enantiomer. In considering why reactivity is unfavorable for (*E*)-enediketones, several potential transitions states derived from methyl or phenyl ketone-coordinated isomers of **6** were examined, but only the transition states **8f** and **8g** converged reliably. Notably, ouroboros activation was not observable for any reasonable geometries corresponding to transition states derived from (*E*)-substrates, the 6-endo-trig transition state was

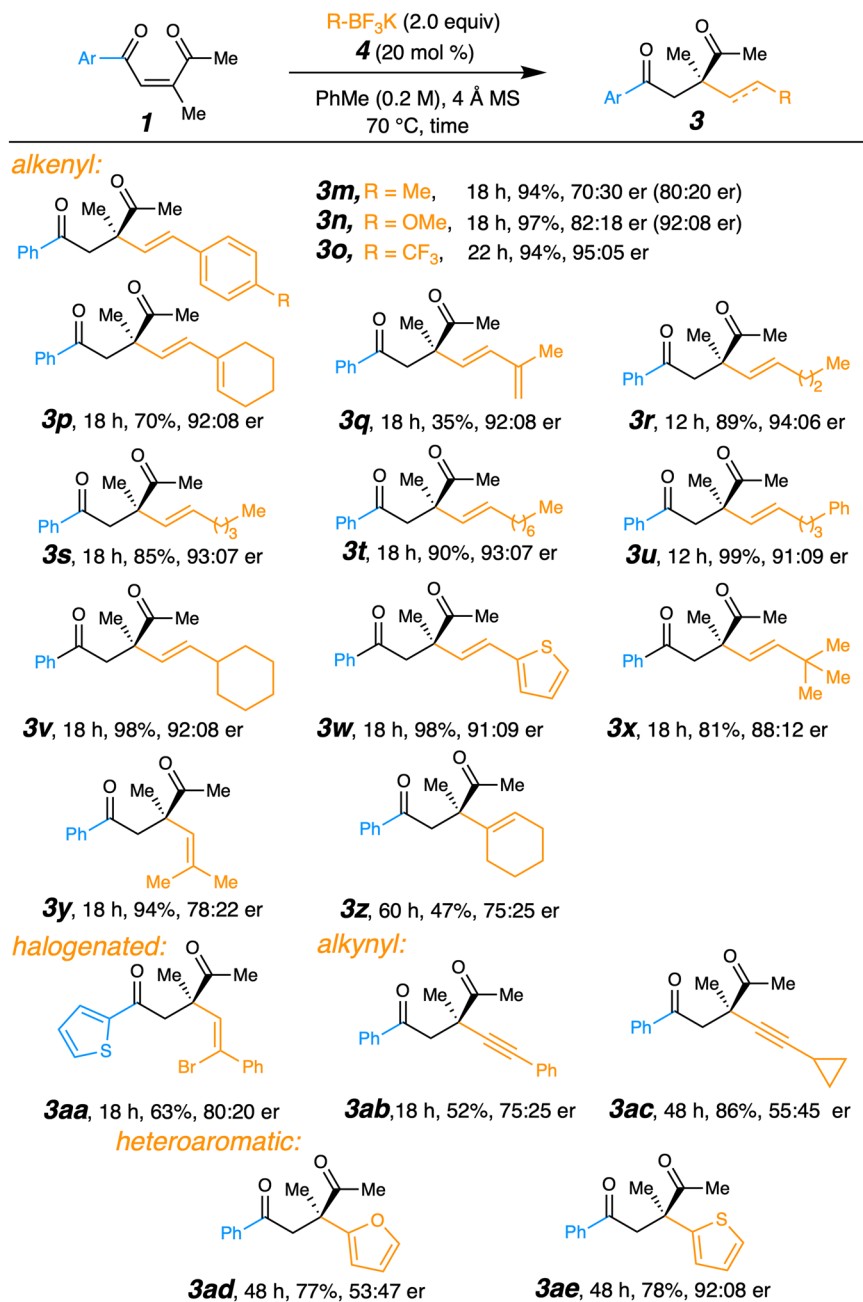

**Fig. 5 | Substrate scope of products from various nucleophiles.** Reaction yields are of purified isolated products, with the average of at least 2 trials presented. Enantiomeric ratios were determined by HPLC with chiral stationary phase. For **3m–n** and **3s–3x**, the use of 40 mol % of **4** is presented. For **3s–x** and **3ae**, the solvent was PhCF₃. For **3aa** the use 30 mol % of **4** allowed reduction of organoborate nucleophile to 1.5 equiv.

thus lower in energy, and the resulting higher overall barrier explains the lack of reactivity of β-disubstituted enones in all previous studies[9]. Given other recent efforts that also observe such a dependence on (Z)-enone geometry[25–28], ouroboros stabilization may be operative in many catalytic reactions. The poor er arising from the lack of rotation about the alkene-ketone bond in forming **3l** also aligns with this hypothesis, as the carbonyl could not fully rotate out of plane to provide 5-exo reactivity, forcing it through a 6-endo transition state like **8e**, which also lacks ouroboros activation and therefore has reduced stereocontrol. The decreased enantioselectivity and altered regioselectivity seen for **5g** and **5h** could also be due to competitive 6-endo reactivity for those substrates rather than due to Lewis basicity.

An examination of the calculated LUMOs in intermediate **6** and its phenylketone-coordinated isomer showed localization on the planar enone (HOMO/LUMO illustrated in the Source Data file), but the carbons undergoing nucleophilic attack (C13 or C14) bear different proportions of the LUMO[74]. For both isomers, C14 has significant LUMO character, leading to better HOMO/LUMO overlap, but less of the LUMO is located on C13. The relative localization of the LUMO in these structures and the relative energies of the subsequent transition states correspond to previously characterized experimental rate dependencies on the stabilization of developing cationic charge at the β-carbon of the enone in this class of conjugate additions[75,76]. An additional insight into the regioselectivity was obtained by examining two possible n→π* interactions in the Z-enediketone. That arising from the Ph-ketone donating into a twisted Me-ketone resulted in a 1.7 kcal/mol more stable conformation than that arising from the Me-ketone donating into a twisted Ph-ketone. The stereoelectronic and steric

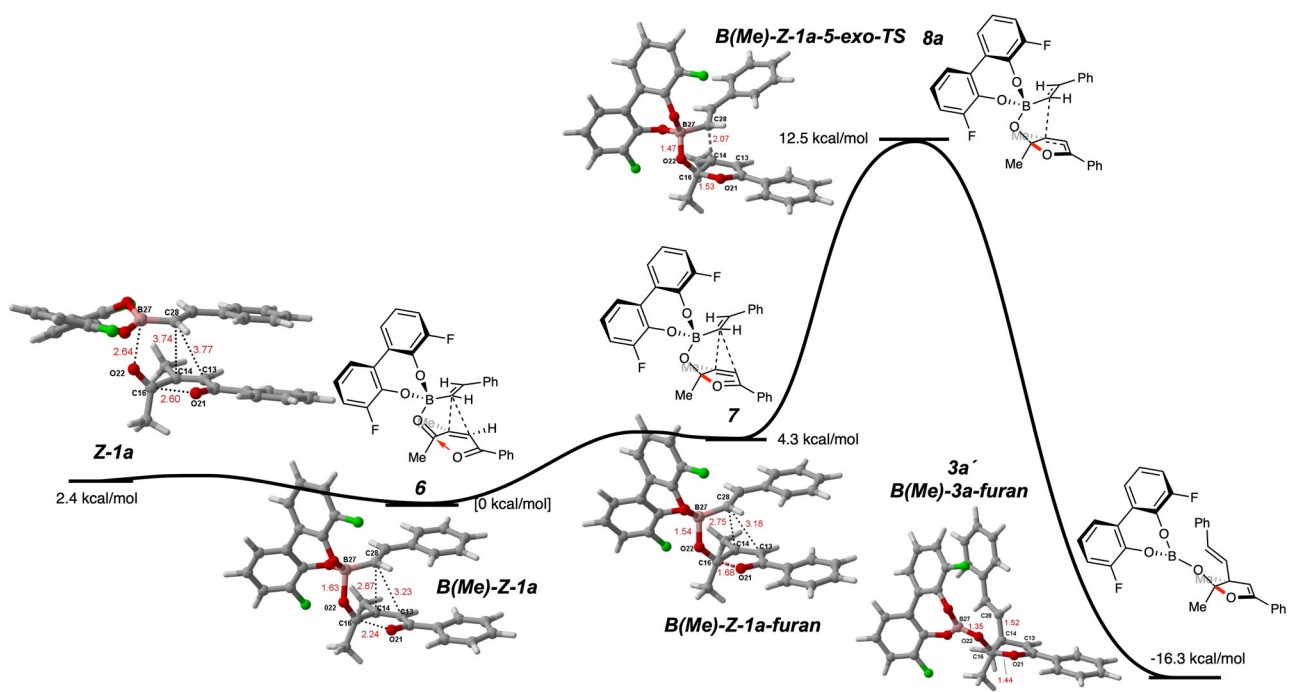

**Fig. 6 | Proposed mechanistic intermediates for lowest energy reaction pathway.** The BINOL catalyst was modelled as 3,3′-difluorobisphenol. The pre-transition state Lewis acid/Lewis base complex was defined as 0 kcal/mol and other optimized structures are reported relative that energy.

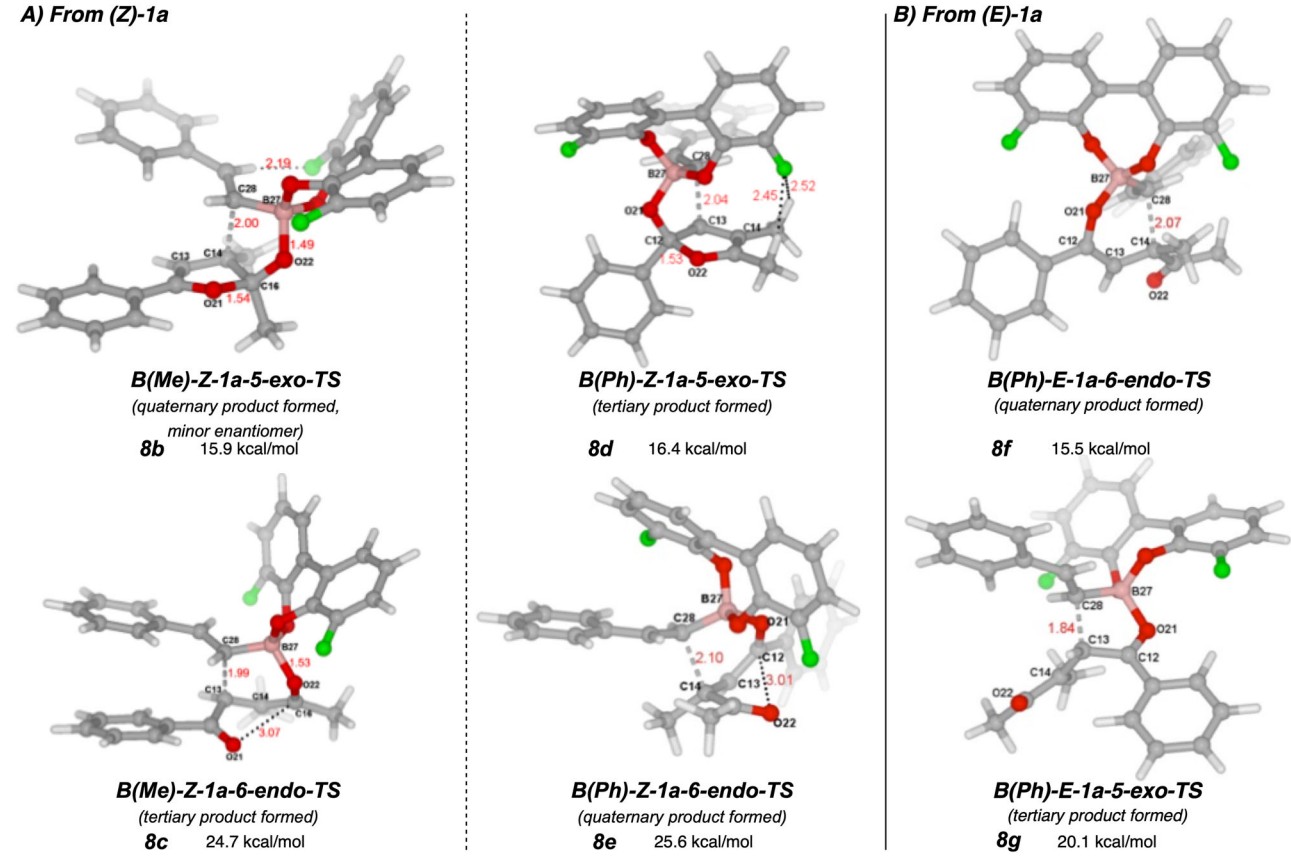

**Fig. 7 | Alternative Transition States with and without ouroboros activation.** Energies are defined relative to **B(Me)-Z-1a (6)** in Fig. 6. **A** Transition state structures derived from (**Z**)−**1a**. **B** Transition state structures derived from (**E**)−**1a**. (Me) and (Ph) define whether the methyl ketone or phenyl ketone are bound by the Lewis acidic boron, respectively. 5-exo and 6-endo define the geometry of C–C bond formation according to Baldwin's rules. **8b** and **8d** show significant ouroboros stabilization.

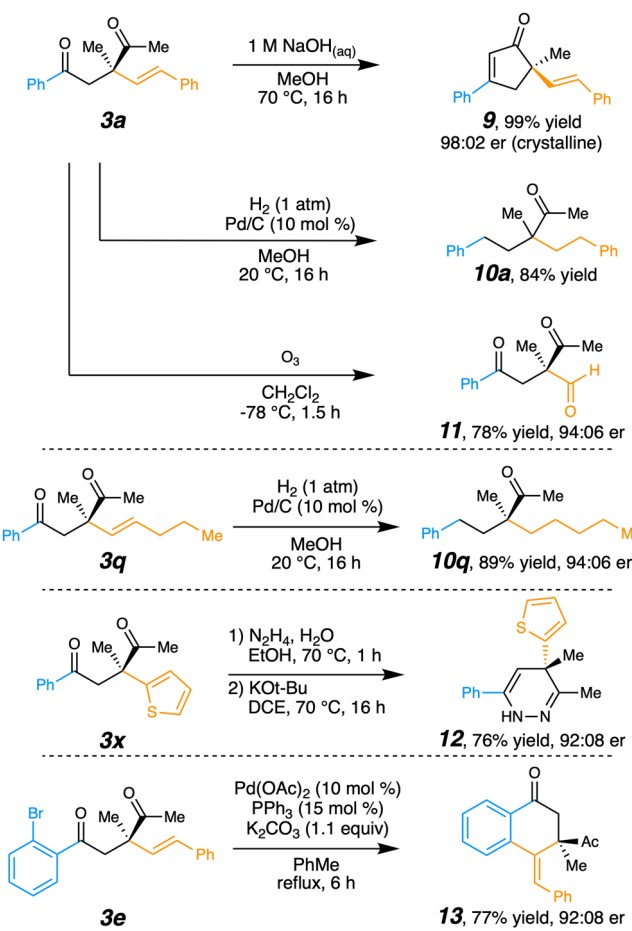

**Fig. 8 | Utility of quaternary diketone products.** The chemical synthesis of several important molecular motifs is illustrated. Examples include α-quaternary cyclopentenes (**9**), quaternary alkanes that are achiral (**10a**) or chiral and enantioenriched (**10q**), tricarbonyls (**11**), dihydropyridazines (**12**), and methide-quinones (**13**).

HOMO/LUMO overlap in the formation of the quaternary carbon relative to the tertiary carbon. A broad substrate scope of chiral α-quaternary 1,4-diketones were synthesized. Further transformations to quaternary carbon-containing enantio-enriched cyclopentenones, linear hydrocarbons, dihydropyridazines, and quinone methides were demonstrated in good yield and er. These building blocks will enable synthetic endeavors in many areas.

## Data availability
The experimental data generated in this study and computational procedures with optimized structures are provided in the Supplementary Information file. The molecular coordinate data generated in this study are provided in the Source Data file. All primary data files, such as .fid files for NMR spectra or coordinate files for molecular structures, are available from the corresponding author for free upon request. The X-ray crystallographic coordinates for structures reported in this study have been deposited at the Cambridge Crystallographic Data Centre (CCDC), under deposition number 2121715 (**9**). These data can be obtained free of charge from The Cambridge Crystallographic Data Centre via www.ccdc.cam.ac.uk/data_request/cif. Source data are provided with this paper.

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

interactions in these conformations are also likely to be present in **8a** and **8d**, and the relative stability of their geometries contributing to the difference in regioisomeric transition state energies.

To demonstrate the utility of these quaternary 1,4-diketone products[58–63,77], examples were transformed into key synthetic building blocks (Fig. 8). Chiral cyclopentenones (see 8), which exist widely in bioactive compounds[78–80], could be formed in high yield and er via aldol condensation. The absolute stereochemistry of **9** was confirmed by X-ray crystallography[81]. Chemoselective hydrogenation reduced the benzoyl carbonyl and the alkene of **3a** or **3q** to form the aliphatic quaternary carbon centers in **10a** and **10q**, respectively, with the latter containing an otherwise difficult to access alkylated quaternary center with high er. The oxidative cleavage of the styrenyl olefin gave ketoaldehyde **11**, which would be useful in recently reported pyrrolidine syntheses[77,82]. Non-planar heterocycle dihydropyridazine **12**, a bioactive pharmacophore[83,84], could be generated in good yield. Using the bromo substrate **3e** to incorporate an intramolecular Heck coupling reaction gave the quinone-like derivative **12**[85].

In conclusion, we successfully synthesized challenging quaternary centers enantioselectively from (Z)-1,4-enediketones via organocatalyzed conjugate addition. Control experiments showed that the cis-relationship of the ketones was vital to reactivity, and keto-ene bond rotation at the location of C–C bond formation was important for enantioselectivity. DFT calculations showed that the additional ketone provided 5-exo-trig reactivity and a stabilizing interaction through an n→π*cyclic ouroboros activation. The regioselectivity for quaternary carbon formation appeared to be based primarily on the greater

15. Wu, J., Mampreian, D. M. & Hoveyda, A. H. Enantioselective synthesis of nitroalkanes bearing all-carbon quaternary stereogenic centers through Cu-catalyzed asymmetric conjugate additions. *J. Am. Chem. Soc.* **127**, 4584–4585 (2005).

16. Dabrowski, J. A., Villaume, M. T. & Hoveyda, A. H. Enantioselective synthesis of quaternary carbon stereogenic centers through copper-catalyzed conjugate additions of aryl- and alkylaluminum reagents to acyclic trisubstituted enones. *Angew. Chem. Int. Ed.* **52**, 8156–8159 (2013).

17. Endo, K., Hamada, D., Yakeishi, S. & Shibata, T. Effect of multi-nuclear copper/aluminum complexes in highly asymmetric conjugate addition of trimethylaluminum to acyclic enones. *Angew. Chem. Int. Ed.* **52**, 606–610 (2013).

18. Fillion, E. & Wilsily, A. Asymmetric synthesis of all-carbon benzylic quaternary stereocenters via cu-catalyzed conjugate addition of dialkylzinc reagents to 5-(1-arylalkylidene) meldrum's acids. *J. Am. Chem. Soc.* **128**, 2774–2775 (2006).

19. Sun, Y., Zhou, Y., Shi, Y., Del Pozo, J., Torker, S. & Hoveyda, A. H. Copper-hydride-catalyzed enantioselective processes with allenyl boronates. Mechanistic nuances, scope, and utility in target-oriented synthesis. *J. Am. Chem. Soc.* **141**, 12087–12099 (2019).

20. Mauleon, P. & Carretero, J. C. Enantioselective construction of stereogenic quaternary centres via Rh-catalyzed asymmetric addition of alkenylboronic acids to α,β-unsaturated pyridylsulfones. *Chem. Commun.* **39**, 4961–4963 (2005).

21. Shintani, R., Tsutsumi, Y., Nagaosa, M., Nishimura, T. & Hayashi, T. Sodium tetraarylborates as effective nucleophiles in rhodium/diene-catalyzed 1,4-addition to β,β-disubstituted α,β-unsaturated ketones: catalytic asymmetric construction of quaternary carbon stereocenters. *J. Am. Chem. Soc.* **131**, 13588–13589 (2009).

22. Oliveira, Vd. G., Cardoso, M. Fd. C. & Forezi, Ld. S. M. Organocatalysis: a brief overview on its evolution and applications. *Catalysts* **8**, 605 (2018).

23. Shaikh, I. R. Organocatalysis: Key Trends in Green Synthetic Chemistry, Challenges, Scope towards Heterogenization, and Importance from Research and Industrial Point of View. *J. Catal.* **2014**, 402860 (2014).

24. Han, B., He, X.-H., Liu, Y.-Q., He, G., Peng, C. & Li, J.-L. Asymmetric organocatalysis: an enabling technology for medicinal chemistry. *Chem. Soc. Rev.* **50**, 1522–1586 (2021).

25. Pierrot, D. & Marek, I. Synthesis of enantioenriched vicinal tertiary and quaternary carbon stereogenic centers within an acyclic chain. *Angew. Chem. Int. Ed.* **59**, 36–49 (2020).

26. Govender, T., Arvidsson, P. I., Maguire, G. E., Kruger, H. G. & Naicker, T. Enantioselective organocatalyzed transformations of β-ketoesters. *Chem. Rev.* **116**, 9375–9437 (2016).

27. Tian, L., Luo, Y.-C., Hu, X.-Q. & Xu, P.-F. Recent developments in the synthesis of chiral compounds with quaternary centers by organocatalytic cascade reactions. *Asian J. Org. Chem.* **5**, 580–607 (2016).

28. Wang, J., He, F. & Yang, X. Asymmetric construction of acyclic quaternary stereocenters via direct enantioselective additions of α-alkynyl ketones to allenamides. *Nat. Commun.* **12**, 6700 (2021).

29. Mukherjee, S., Yang, J. W., Hoffmann, S. & List, B. Asymmetric enamine catalysis. *Chem. Rev.* **107**, 5471–5569 (2007).

30. Tsogoeva, S. B. Recent advances in asymmetric organocatalytic 1,4-conjugate additions. *Eur. J. Org. Chem.* **2007**, 1701–1716 (2007).

31. Nguyen, T. N. & May, J. A. Enantioselective organocatalytic conjugate addition of organoboron nucleophiles. *Tetrahedron Lett.* **58**, 1535–1544 (2017).

32. Shim, J. H., Cheun, S. H., Kim, H. S. & Ha, D.-C. Organocatalysis for the asymmetric michael addition of aldehydes and α,β-unsaturated nitroalkenes. *Catalysts* **12**, 121 (2022).

33. Melchiorre, P. & Jørgensen, K. A. Direct enantioselective michael addition of aldehydes to vinyl ketones catalyzed by chiral amines. *J. Org. Chem.* **68**, 4151–4157 (2003).

34. Wang, Z., Yang, Z. P. & Fu, G. C. Quaternary stereocentres via catalytic enantioconvergent nucleophilic substitution reactions of tertiary alkyl halides. *Nat. Chem.* **13**, 236–242 (2021).

35. Bella, M. & Gasperi, T. Organocatalytic formation of quaternary stereocenters. *Synthesis* **2009**, 1583–1614 (2009).

36. Alam, R., Vollgraff, T., Eriksson, L. & Szabó, K. J. Synthesis of adjacent quaternary stereocenters by catalytic asymmetric allylboration. *J. Am. Chem. Soc.* **137**, 11262–11265 (2015).

37. Wheatley, E., Zanghi, J. M. & Meek, S. J. Diastereo-, enantio-, and anti-selective formation of secondary alcohol and quaternary carbon stereocenters by cu-catalyzed additions of B-substituted allyl nucleophiles to carbonyls. *Org. Lett.* **22**, 9269–9275 (2020).

38. Chen, L., Pu, M., Li, S., Sang, X., Liu, X., Wu, Y.-D. & Feng, X. Enantioselective synthesis of nitriles containing a quaternary carbon center by michael reactions of silyl ketene imines with 1-acrylpyrazoles. *J. Am. Chem. Soc.* **143**, 19091–19098 (2021).

39. Akagawa, K. & Kudo, K. Construction of an all-carbon quaternary stereocenter by the peptide-catalyzed asymmetric michael addition of nitromethane to β-Disubstituted α,β-Unsaturated Aldehydes. *Angew. Chem. Int. Ed.* **51**, 12786–12789 (2012).

40. Kawai, H., Yuan, Z., Kitayama, T., Tokunaga, E. & Shibata, N. Efficient access to trifluoromethyl diarylpyrrolines and their n-oxides through enantioselective conjugate addition of nitromethane to β,β-Disubstituted Enones. *Angew. Chem. Int. Ed.* **52**, 5575–5579 (2013).

41. Kawai, H., Okusu, S., Tokunaga, E., Sato, H., Shiro, M. & Shibata, N. Organocatalytic asymmetric synthesis of trifluoromethyl-substituted diarylpyrrolines: enantioselective conjugate cyanation of β-Aryl-β-trifluoromethyl-disubstituted Enones. *Angew. Chem. Int. Ed.* **51**, 4959–4962 (2012).

42. Kwiatkowski, P., Cholewiak, A. & Kasztelan, A. Efficient and highly enantioselective construction of trifluoromethylated quaternary stereogenic centers via high-pressure mediated organocatalytic conjugate addition of nitromethane to β,β-Disubstituted Enones. *Org. Lett.* **16**, 5930–5933 (2014).

43. Wu, T. R. & Chong, J. M. Asymmetric conjugate alkenylation of enones catalyzed by chiral diols. *J. Am. Chem. Soc.* **129**, 4908–4909 (2007).

44. Paton, R. S., Goodman, J. M. & Pellegrinet, S. C. Theoretical study of the asymmetric conjugate alkenylation of enones catalyzed by binaphthols. *J. Org. Chem.* **73**, 5078–5089 (2008).

45. Lou, S., Moquist, P. N. & Schaus, S. E. Asymmetric allylboration of acyl imines catalyzed by chiral diols. *J. Am. Chem. Soc.* **129**, 15398–15404 (2007).

46. Bishop, J. A., Lou, S. & Schaus, S. E. Enantioselective addition of boronates to acyl imines catalyzed by chiral biphenols. *Angew. Chem. Int. Ed.* **48**, 4337–4340 (2009).

47. Shih, J.-L., Nguyen, T. S. & May, J. A. Organocatalyzed asymmet-ric conjugate addition of heteroaryl and aryl trifluoroborates: a synthetic strategy for discoipyrrole D. *Angew. Chem. Int. Ed.* **54**, 9931–9935 (2015).

48. Nguyen, T. N., Chen, P.-A., Setthakarn, K. & May, J. A. Chiral diol-based organocatalysts in enantioselective reactions. *Molecules* **23**, 2317 (2018).

49. Sundstrom, S., Nguyen, T. S. & May, J. A. Relay catalysis to synthesize β-substituted enones: organocatalytic substitution of vinylogous esters and amides with organoboronates. *Org. Lett.* **22**, 1355–1359 (2020).

50. Peng, P.-K. & May, J. A. Enantioselective organocatalytic conjugate addition in a tandem synthesis of δ-substituted cyclohexenones and four-step total synthesis of penienone. *Org. Lett.* **24**, 5334–5338 (2022).

51. Yao, E.-Z., Chai, G.-L., Zhang, P., Zhu, B. & Chang, J. Chiral dihydroxytetraphenylene-catalyzed enantioselective conjugate addition of boronic acids to β-enaminones. *Org. Chem. Front.* **9**, 2375–2381 (2022).

52. Wang, X., Chai, G.-L., Hou, Y.-J., Zhou, M.-Q. & Chang, J. Enantio-selective synthesis of chiral organosilicon compounds by organo-catalytic asymmetric conjugate addition of boronic acids to β-Silyl-α,β-Unsaturated. *Ketones. J. Org. Chem.* **88**, 3254–3265 (2023).

53. Chai, G.-L., Zhang, P., Yao, E.-Z. & Chang, J. Enantioselective conjugate addition of boronic acids to α,β-Unsaturated 2-Acyl imidazoles catalyzed by chiral diols. *J. Org. Chem.* **87**, 9197–9209 (2022).

54. Chai, G.-L., Sun, A. Q., Zhai, D., Wang, J., Deng, W.-Q., Wong, H. N. C. & Chang, J. Chiral hydroxytetraphenylene-catalyzed asymmetric conjugate addition of boronic acids to enones. *Org. Lett.* **21**, 5040–5045 (2019).

55. Meyers, A. I., Harre, M. & Garland, R. Asymmetric synthesis of quaternary carbon centers. *J. Am. Chem. Soc.* **106**, 1146–1148 (1984).

56. Zhang, H., Wang, Z., Wang, Z., Chu, Y., Wang, S. & Hui, X.-P. Visible-light-mediated formal carbene insertion reaction: enantioselective synthesis of 1,4-dicarbonyl compounds containing all-carbon quaternary stereocenter. *ACS Catal.* **12**, 5510–5516 (2022).

57. Kaldre, D., Klose, I. & Maulide, N. Stereodivergent synthesis of 1,4-dicarbonyls by traceless charge– accelerated sulfonium re-arrangement. *Science* **361**, 664–667 (2018).

58. Fujimoto, T., Endo, K., Tsuji, H., Nakamura, M. & Nakamura, E. Construction of a chiral quaternary carbon center by indium-catalyzed asymmetric α-Alkenylation of β-Ketoesters. *J. Am. Chem. Soc.* **130**, 4492–4496 (2008).

59. Bella, M. & Jørgensen, K. A. Organocatalytic enantioselective conjugate addition to alkynones. *J. Am. Chem. Soc.* **126**, 5672–5673 (2004).

60. Corkey, B. K. & Toste, F. D. Catalytic enantioselective Conia-Ene reaction. *J. Am. Chem. Soc.* **127**, 17168–17169 (2005).

61. Gao, F., McGrath, K. P., Lee, Y. & Hoveyda, A. H. Synthesis of quaternary carbon stereogenic centers through enantioselective cu-catalyzed allylic substitutions with vinylaluminum. *Reag. J. Am. Chem. Soc.* **132**, 14315–14320 (2010).

62. Ren, Y., Lu, S., He, L., Zhao, Z. & Li, S.-W. Catalytic asymmetric decarboxylative michael addition to construct an all-carbon quaternary center with 3-Alkenyl-oxindoles. *Org. Lett.* **24**, 2585–2589 (2022).

63. McGrath, K. P., Hoveyda, A. H. & Multicomponent, A. Ni-, Zr-, and Cu-catalyzed strategy for enantioselective synthesis of alkenyl-substituted quaternary carbons. *Angew. Chem. Int. Ed.* **53**, 1910–1914 (2014).

64. Xu, K., Fang, Y., Yan, Z., Zha, Z. & Wang, Z. A highly tunable stereoselective dimerization of methyl ketone: efficient synthesis of *E*- and *Z*–1,4-Enediones. *Org. Lett.* **15**, 2148–2151 (2013).

65. Barbato, K. S., Luan, Y., Ramella, D., Panek, J. S. & Schaus, S. E. Enantioselective multicomponent condensation reactions of phenols, aldehydes, and boronates catalyzed by chiral Bi-phenols. *Org. Lett.* **17**, 5812–5815 (2015).

66. Baldwin, J. E. Rules for ring closure. *J.C.S. Chem. Comm.* **18**, 734–736 (1976).

67. Baldwin, J. E., Thomas, R. C., Kruse, L. I. & Silberman, L. Rules for ring closure: ring formation by conjugate addition of oxygen nucleophiles. *J. Org. Chem.* **42**, 3846–3852 (1977).

68. Roscales, S. & Csáký, A. G. Transition-metal free reactions of boronic acids: cascade addition – ring-opening of furans towards functionalized γ-ketoaldehydes. *Chem. Commun.* **52**, 3018–3021 (2016).

69. Sahariah, B. & Sarma, B. K. Relative orientation of the carbonyl groups determines the nature of orbital interactions in carbonyl–carbonyl short contacts. *Chem. Sci.* **10**, 909–917 (2019).

70. Donald, K. J., Gillespie, S. & Shafi, Z. Ouroboros: Heterocycles closed by dative σ bonds and stabilized by π delocalization. *Tetrahedron* **75**, 335–345 (2019).

71. Maity, R., Ghosh, S. & Das, I. Integrating regioselective E→Z iso-merization of trienones with cascade sequences under photosensitizer-free direct irradiation at 390|nm. *Eur. J. Chem.* **29**, e202300421 (2023).

72. Newberry, R. W. & Raines, R. T. The n→π* Interaction. *Acc. Chem. Res.* **50**, 1838–1846 (2017).

73. Vik, E. C., Li, P., Pellechia, P. J. & Shimizu, K. D. Transition-state stabilization by n→π* interactions measured using molecular rotors. *J. Am. Chem. Soc.* **141**, 16579–16583 (2019).

74. See computed geometry coordinates for B(Me)-*Z*–1a (**6a**) and B(Ph)-*Z*–1a (**6b**) in the supporting information, where the LUMO for each is depicted.

75. Nguyen, T. S., Yang, M. S. & May, J. A. Experimental mechanistic insight into the BINOL-catalyzed enantioselective conjugate addition of boronates to enones. *Tetrahedron Lett.* **56**, 3337–3341 (2015).

76. Brooks, B., Hiller, N. & May, J. A. Reaction rate differences between organotrifluoroborates and boronic acids in BINOL-catalyzed conjugate addition to enones. *Tetrahedron Lett.* **83**, 153412 (2021).

77. Clift, M. D., Taylor, C. N. & Thomson, R. J. Oxidative carbon–carbon bond formation via silyl bis-enol ethers: controlled cross-coupling for the synthesis of quaternary centers. *Org. Lett.* **9**, 4667–4669 (2007).

78. Roberts, S. M., Santoro, M. G. & Sickle, E. S. The emergence of the cyclopentenone prostaglandins as important, biologically active compounds. *J. Chem. Soc., Perkin Trans.* **1**, 1735–1742 (2002).

79. Rezgui, F., Amri, H. & El Gaïed, M. M. Synthetic methods for α-substituted cyclic α,β-enones. *Tetrahedron* **59**, 1369–1380 (2003).

80. Lee, Y. H., Denton, E. H. & Morandi, B. Modular cyclopentenone synthesis through the catalytic molecular shuffling of unsaturated acid chlorides and alkynes. *J. Am. Chem. Soc.* **142**, 20948–20955 (2022).

81. Structure data for crystal of cyclopentenone 9 deposited in the Cambridge Crystallographic Data Centre database. CCDC deposition number 2121715.

82. Barløse, C. L., Østergaard, N. L., Bitsch, R. S., Iversen, M. V. & Jørgensen, K. A. A direct organocatalytic enantioselective route to functionalized trans-diels–alder products having the norcarane scaffold. *Angew. Chem. Int. Ed.* **60**, 18318–18327 (2021).

83. Wermuth, C. G. Are pyridazines privileged structures? *MedChemComm* **2**, 935–941 (2011).

84. Lovering, F., Bikker, J. & Humblet, C. Escape from flatland: increasing saturation as an approach to improving clinical success. *J. Med. Chem.* **52**, 6752–6756 (2009).

85. Machotta, A. B., Straub, B. F. & Oestreich, M. Oxygen donor-mediated equilibration of diastereomeric alkene–palladium(II) intermediates in enantioselective desymmetrizing heck cyclizations. *J. Am. Chem. Soc.* **129**, 13455–13463 (2007).

86. Tanaka, N., Kakuguchi, Y., Ishiyama, H., Kubota, T. & Kobayashi, J. I. Yezo'otogirins A–C, new tricyclic terpenoids from Hypericum yezoense. *Tetrahedron Lett.* **50**, 4747–4750 (2009).

## Acknowledgements

The authors are grateful for generous financial support from the NSF (grant CHE-2102282, JAM) and the Welch Foundation (grant E–1744, JAM). Professor Judy Wu at the University of Houston is thanked for aid in access to computational facilities. The computational work was completed in part with resources provided by the Research Computing Data Core at the University of Houston. We are grateful to Dr. Sasha Sundstrom and AbbVie North Chicago for aiding in the acquisition of specific rotation data.

## Author contributions

P.-K.P.: Conceptualization, experimental methodology development and investigation, validation of structures, writing of first draft. A.I.: computational model development and investigation, validation of transitiona states and intermediates, and electronic data curation. J.A.M.: conceptualization of project, writing and editing, project supervision, and acquisition of funding.

## Competing interests

The authors declare no competing interests.
