## [Peer Review File · Nature Communications]

REVIEWER COMMENTS

Reviewer #1 (Remarks to the Author):

In this manuscript the regio- and enantioselective conjugate addition of organoboron compounds to acyclic enediketones catalyzed by chiral diols is described. The reaction proceeds in a highly regioselective fashion to give the β -substituted 1,4-dicarbonyl product having a quaternary stereocenter with very good to excellent yields and enantiomeric ratios in most cases. Potassium trifluoroorganoborate nucleophiles together with chiral 3,3'-bis(perfluorotoluidyl)-BINOL were found to give optimized yields and enantioselectivities. The reaction scope was investigated relative to the enediketone and the trifluoroorganoborate (mostly alkenyl and a few alkynyl and heteroaryl). In addition, the reaction mechanism was investigated with experiments and DFT calculations. The (Z)-isomer of the ketone was found to be the reactive form, and a $n \rightarrow \pi^*$ donation by the unbound ketone's oxygen lone pair to the boronate-coordinated ketone in a 5-exo trig cyclic transition structure (TS) was proposed to explain the experimentally observed reactivity and selectivity. Finally, the utility of the conjugate addition products was demonstrated by performing a number of synthetic transformations to give different building blocks bearing quaternary carbons. The studied reaction is very interesting, expands the scope of organocatalytic enantioselective conjugate additions to enones and provides a new method to generate quaternary carbon stereogenic centers in cyclic chains. The work is well organized and the conclusions are supported by the experimental data. In my opinion, the manuscript should be accepted for publication, but the following questions should be addressed first.

1. When the different boron sources were examined the potassium trifluoroborate was found to be the best nucleophile. However, it is surprising that the boronic acid reacts much faster but gives a lower *er* (page S-16). Do the authors have an explanation for this observation?
2. The computed free energies of activation shown in the reaction coordinate (Figure 4) are very low. Given such low figures, if the conjugate addition was the rate-determining step, these reactions could be performed at low temperatures, but the reactions are run at 70 °C (page S-15). This might suggest that the exchange of ligands is the more energetically demanding step in the catalytic cycle and could be related to the higher reactivity of the boronic acid?
3. I found the mechanistic proposal very interesting and the $n \rightarrow \pi^*$ donation in the enediketone intriguing. Such stabilizing interaction appears to be operative (but less effective) in the starting uncomplexed (Z)-enediketone, as suggested by the structure for (Z)-1a shown in the Supporting Information (S-173). Is this the lowest energy conformation? Is the $n \rightarrow \pi^*$ donation from the methyl ketone to the phenyl ketone less stabilizing, and/or the corresponding alternative conformation less

stabilized by conjugation? Does NMR data provide any hint? This might help clarify the origin of regioselectivity.

4. The numbering and the drawing of some structures in Figure 4 and the ones shown in the Supporting Information are a bit confusing. For instance, in Figure 4 the complexes seem to involve the wrong lone pair of the enone for $n \rightarrow \pi^*$ donation but the geometries given in the Supporting Information seem to be fine.

5. Inspection of the most favorable TSs (7a and 7b) should give some insight to account for the observed enantioselectivity. Is there a clash between the BINOL substituent (fluorine in the simplified model) and a vinyl proton of the styryl moiety in 7b?

6. Apparently, the authors have run NBO and FMO calculations (lines 156-162 and 382-383, ref. 76), but such results are not shown. I suggest changes on LUMO energies for the enediketone after complexation, and also the computed energy for $n \rightarrow \pi^*$ donation, are reported.

7. Was the reaction coordinate for the competing background reaction investigated theoretically?

Minor comments:

Line 10: In the first sentence of the abstract the words “vinylated or arylated” should be replaced by “alkenylated, alkynylated and heteroarylated”.

Line 82: benzylic should be replaced by phenyl.

Line 84: Pelligrinet is misspelled (correct spelling Pellegrinet).

Reviewer #2 (Remarks to the Author):

As a computational organic chemist I evaluated the computational study only. The results are very valuable and quite well presented and the manuscript deserves the publication after a revision because there are some points that makes me quite puzzled:

1. As far I can see from the figures in the SI in TS 7c, 7c' , 7e , 7f, 7f' , and 7g the carbonyl oxygen atoms(those not bound to the Boron) are quite far from the opposite carbonyl carbon atoms. In some cases (7c' , 7e ,7f' and 7g) the oxygens seem to be even in the wrong position.

So, how these TSs can connect to the enolacetates?

1b. It would be also useful to report these distances in all TSs figures in the SI).

2. By contrast, in the TSs 7a, 7b, 7d, and 7d' these carbon-oxygen distances are very short (ca 1.5 Å) suggesting that in the starting poin of these TSs the C-O bonds are already formed, i.e. there are some intermediates between the intermediates 6 and the TSs 7. As an alternative, the cites TSs can be very "late" from the point of view of the formation of the C-O bonds. The authors should explore these alternative.

But the real question is: where the IRCs really performed from all TSs?

3. It would be also usefull to report the C12-O22 distances in 6b and 6b' and C16-O21 in 6a , 6a' and 6a".

Other main points:

a. Part of the discussion of the computational data is base on he shape of some HOMOs and LUMOs. Authors must report the relative figures at least in the SI.

b. In Figure 5 there are some errors:

* 3a': for homogeneity with the TS in C16 and C14 there shoul be indicated the Me

* 5a': In C16 there should be a Me, not a Ph

Reviewer #3 (Remarks to the Author):

Enantioselective 1,4-addition of boric acid or derivatives with Michael receptors catalyzed by chiral BINOLs has proven to be a useful reaction to produce chiral tertiary carbon centers. However, chiral acyclic all-carbon quaternary stereocenters have never been synthesized by this reaction. The manuscript by May et al. concerns the enantioselective formation of chiral 1,4-diketones containing acyclic all-carbon quaternary stereocenters by a 1,4-conjugate addition. Disubstituted enediketones and potassium styrenyl trifluoroborates were catalyzed by 3,3'-bis(perfluorotoluy)l-BINOL (Figures 2–3). It is suggested that only (Z)-enediketones performed the 1,4-addition, and the E-isomers were almost completely unreactive. This reaction gave good regioselectivities. The reasons for this were revealed by experiments with DFT calculations (Figure 4). However, the enantioselectivities of this work are currently not well resolved. Only five products had $\geq 90\%$ ee. Although the authors tried 31 examples, the enantioselectivities of more than 1/3 of the products (12 examples) were below 80% ee. Some examples (3j, 3m, 3n, 3q, 3s, 3t, 3u, 3w, 3x, 9 examples) require 40 mol% catalyst. The amount of catalyst is too much, and the authors should consider this point. The authors should re-optimize the conditions to further improve the stereoselectivities of the products. The authors performed DFT calculations that explain the regioselectivity of the reaction well, but do not reveal the effect of the catalyst substituents or the reaction substrate on the enantioselectivity. The value of the synthetic application of the method was not shown very well. It is suggested that the authors apply the methodology to the synthesis of natural products or drug molecule analogues (Figure 1) to better demonstrate the value of the methodology.

In summary, the quality of the manuscript currently does not meet the publication standards of Nature Communication. I recommend publication of this manuscript in a more specialized journal.

Minor revision:

Manuscript:

(1) Line 74: "Heteroaromatic enone substituents were likewise accommodated (see 3f)." There is only one example where the singular form should be used.

(2) Compound numbers should be bolded, please check the full text.

(3) Line: 119-120: "4" should be bolded, "PhCF₃" should be subscripted.

(4) Line 167: The structure of compound 5a' in figure 4 may be wrong.

(5) Line 176: 3p should be changed to 3q.

(6) Line 186 (Z)-1,4-enediketones "Z" should be italicized, please check the full text.

References:

(7) The format of the references is not standardized according to NC Guideline. Some errors in the format of the references.

Ref. 3, 4, 12, 13, 19, 22-33, 35, 36, 37, 49, 50, 62, 66-73, 72, 73, 76.

Supporting Information

(8) Data of the HPLC: retention time should be pointed.

(9) ¹⁹F NMR spectra of 1d, 1o, 3d and 3o should be provided.

(10) The X-ray crystallography of 8 is missing in SI.

(11) Specific rotation data of the products are not available.

(12) The yields of 3j, 3l, 3r, 3y in SI are different from the yields in the manuscript. The authors should check the yields of all products.

(13) There are some impurities appeared in the NMR spectra, it is recommended to do further purification: 1b, 3n, 3aa.

RESPONSE TO REVIEWER COMMENTS

Reviewer #1:

1. When the different boron sources were examined the potassium trifluoroborate was found to be the best nucleophile. However, it is surprising that the boronic acid reacts much faster but gives a lower e_r (page S-16). Do the authors have an explanation for this observation?

RESPONSE:

We observed a racemic background reaction in many cases, possibly facilitated by the Brønsted acidity of the Boronic acid. The rapidity of reaction for the boronic acids could provide a faster uncatalyzed background reaction, which lowers the e_r observed. To confirm this hypothesis, we have run additional control experiments without catalysts to compare rates.

2. The computed free energies of activation shown in the reaction coordinate (Figure 4) are very low. Given such low figures, if the conjugate addition was the rate-determining step, these reactions could be performed at low temperatures, but the reactions are run at 70 °C (page S-15). This might suggest that the exchange of ligands is the more energetically demanding step in the catalytic cycle and could be related to the higher reactivity of the boronic acid?

RESPONSE:

The reviewer highlights an important point about the absolute value of the energies involved. We believe the low absolute values for the energies involved are a result of the level of theory used and that we reference a “0” energy point that is already on the reaction pathway rather than the starting materials, so the absolute energy is likely higher. We have re-examined the methods and calculations, and we observed minor energy barriers for both complexation and furan intermediate formation, suggesting that these species exist in equilibria. We do not believe ligand exchange to be the rate-determining step in the reaction, having also performed a variety of reaction rate experiments in many transformations that show a first-order dependence on the electrophile as well as electrophilic Hammett correlations that would not be operative if ligand exchange with the boronate were rate-limiting. Nevertheless, the reviewer’s insight is significant, and ligand exchange has not been extensively investigated computationally. We will begin such an analysis.

We did try room temperature reactions using trifluoroboronate salts, and the reaction proceeded but required more than 4 days. One potential reason could be that the low solubility of trifluoroboronates, especially at lower temperatures, can suppress the dissolved concentration and slow the reactivity. Low solubility of the trifluoroborate could also cause a slower rate relative to the boronic acid.

To address the comment on the higher rate of the boronic acid, calculations with boronic acids (i.e., without the catalyst) were higher in energy, but the results did not pass IRC checks and so were not included or fully trusted. We have been re-examining these efforts for boronic acids, difluoroboronates, and other organoboron species as nucleophiles. In the meantime, experimental controls show that there is a rapid uncatalyzed background reaction for boronic acids, likely promoted by their Brønsted acidity. Calculated pathways with boronic acids thus likely need to take such modes of action into account.

Another aspect for this work is that the full catalyst is not represented in the calculations performed. In preliminary examination with the fully elaborated ligand, we see indications that

the overall energies will be higher for the catalyzed reaction. However, these calculations are highly time-consuming and will not be ready until likely next year.

3. I found the mechanistic proposal very interesting and the $n \rightarrow \pi^*$ donation in the enediketone intriguing. Such stabilizing interaction appears to be operative (but less effective) in the starting uncomplexed (Z)-enediketone, as suggested by the structure for (Z)-1a shown in the Supporting Information (S-173). Is this the lowest energy conformation? Is the $n \rightarrow \pi^*$ donation from the methyl ketone to the phenyl ketone less stabilizing, and/or the corresponding alternative conformation less stabilized by conjugation? Does NMR data provide any hint? This might help clarify the origin of regioselectivity.

RESPONSE:

The reviewer makes another observant insight here. We did perform a conformation search on the starting diketones, and the illustrated conformation exhibits the lowest energy. The comparable conformation with the $n \rightarrow \pi^*$ donation from the methyl ketone to the phenyl ketone is less stabilizing, as the conformation is 1.7 kcal/mol higher in energy. The relative energy difference thus observed does parallel that observed in the computed transitions states and may be a component of that difference in that geometries are similar. Low temperature NMR experiments have not yet been performed, but this is an excellent idea for our follow-up investigations to obtain experimental evidence for the theoretical intermediates.

We thank the author for the insight, and we will include this possibility in the description of reasons for regioselectivity.

4. The numbering and the drawing of some structures in Figure 4 and the ones shown in the Supporting Information are a bit confusing. For instance, in Figure 4 the complexes seem to involve the wrong lone pair of the enone for $n \rightarrow \pi^*$ donation but the geometries given in the Supporting Information seem to be fine.

RESPONSE:

We thank the reviewer for pointing out this confusion, and we have worked to clarify the depiction. In fact, we have split it into two figures to include new data obtained in response to reviewers' comments. We have also included figures in the SI with individual pathways to various transition states.

5. Inspection of the most favorable TSs (7a and 7b) should give some insight to account for the observed enantioselectivity. Is there a clash between the BINOL substituent (fluorine in the simplified model) and a vinyl proton of the styryl moiety in 7b?

RESPONSE:

With only a 3.4 kcal/mol difference in TS energies, it is difficult to point to a single low-energy interaction to be the cause of the enantioselectivity. In these structures, there is a 2.1 Å distance between the fluorine and the vinyl proton, so the proximity does appear to potentially have an effect. However, we are hesitant to fully ascribe the enantioinduction to this interaction since current calculations with the fully elaborated catalyst that are still in the preliminary stages indicate that this distance is likely to be significantly greater to be more like 2.5 or 2.7 Å. These calculations are significantly slower and more expensive, and so they will be ongoing for a while.

6. Apparently, the authors have run NBO and FMO calculations (lines 156-162 and 382-383, ref. 76), but such results are not shown. I suggest changes on LUMO energies for the enediketone after complexation, and also the computed energy for $n \rightarrow \pi^*$ donation, are reported.

RESPONSE:

We have included these data in the Supplementary Information.

7. Was the reaction coordinate for the competing background reaction investigated theoretically?

RESPONSE:

Calculations with boronic acids (i.e., without the catalyst) were higher in energy, but the results did not pass IRC checks and so were not included. We will be re-examining these efforts for both boronic acids and difluoroboronates as nucleophiles and report the outcomes when we publish the full account of our efforts.

Minor comments:

Line 10: In the first sentence of the abstract the words “vinylylated or arylated” should be replaced by “alkenylated, alkynylated and heteroarylated”.

RESPONSE:

We have made the suggested change.

Line 82: benzylic should be replaced by phenyl.

RESPONSE:

We have made the suggested change.

Line 84: Pelligrinet is misspelled (correct spelling Pellegrinet).

RESPONSE:

We thank the reviewer and have made the suggested correction.

Reviewer #2:

1. As far I can see from the figures in the SI in TS 7c, 7c', 7e, 7f, 7f', and 7g the carbonyl oxygen atoms(those not bound to the Boron) are quite far from the opposite carbonyl carbon atoms. In some cases (7c', 7e, 7f' and 7g) the oxygens seem to be even in the wrong position. So, how these TSs can connect to the enolacetates?

1b. It would be also useful to report these distances in all TSs figures in the SI).

RESPONSE:

We agree that in these structures, no n to π^* interactions are likely, and this is confirmed by NBO analysis. We believe this is why they are higher in energy. We have added C to O bond distances to clarify this in the figures and in the Supplementary Information. We have also attempted to clarify the writing to better present our argument.

2. By contrast, in the TSs 7a, 7b, 7d, and 7d' these carbon-oxygen distances are very short (ca 1.5 Å) suggesting that in the starting point of these TSs the C-O bonds are already formed, i.e. there are some intermediates between the intermediates 6 and the TSs 7. As an alternative, the cited TSs can be very "late" from the point of view of the formation of the C-O bonds. The authors should explore these alternatives.

But the real question is: where the IRCs really performed from all TSs?

RESPONSE:

This is an excellent suggestion, and we are grateful to the reviewer for making it. We have examined the IRCs we made to find localized "shoulders" or transition states. We did indeed find some cases where the n to pi* interaction to generate a furan acetal occurred before the Transition State for C-C bond formation. Each of these was low enough in energy that a cyclic/acyclic equilibrium was likely. For other Transition State trajectories, though, such a shoulder or discrete localized TS was not observed. We have added this information to our discussion and Supplementary Information.

3. It would be also useful to report the C12-O22 distances in 6b and 6b' and C16-O21 in 6a, 6a' and 6a''.

RESPONSE:

This is a good idea, and we have made the change.

Other main points:

a. Part of the discussion of the computational data is based on the shape of some HOMOs and LUMOs. Authors must report the relative figures at least in the SI.

RESPONSE:

We have included these figures in the SI.

b. In Figure 5 there are some errors:

* 3a': for homogeneity with the TS in C16 and C14 there should be indicated the Me * 5a': In C16 there should be a Me, not a Ph

RESPONSE:

We appreciate the reviewer pointing out the error, and we have corrected it.

Reviewer #3:

--The reasons for this were revealed by experiments with DFT calculations (Figure 4). However, the enantioselectivities of this work are currently not well resolved. Only five products had $\geq 90\%$ ee. Although the authors tried 31 examples, the enantioselectivities of more than 1/3 of the products (12 examples) were below 80% ee. Some examples (3j, 3m, 3n, 3q, 3s, 3t, 3u, 3w, 3x, 9 examples) require 40 mol% catalyst. The amount of catalyst is too much, and the authors should consider this point. The authors should re-optimize the conditions to further improve the stereoselectivities of the products.

RESPONSE:

We address this comment in two ways. First, we note that the organocatalyst is highly recyclable. We typically recover ~90% of the catalyst from chromatographic purification, and it can be reused with no loss in efficacy. We will add this fact and the experimental data for its implementation to the Supplementary Information. Secondly, we have demonstrated in previous studies that reactions with this catalyst can be optimized for a single substrate to 5% loading with high yields, but those conditions are typically not general for other substrates. This would pose a significant issue for those trying these conditions for a novel substrate and would require extensive optimization for all substrates we present. We want those trying new substrates to have best chance of success, so we publish the most robust conditions that work generally for the greatest variety of substrates

--The authors performed DFT calculations that explain the regioselectivity of the reaction well, but do not reveal the effect of the catalyst substituents or the reaction substrate on the enantioselectivity.

RESPONSE:

As stated above, we are examining the fully elaborated catalyst in these TSs to obtain a more detailed difference. With only a 2 kcal/mol difference between TS energies, it is difficult to point to definitive interactions. We do note that we have a collaborative paper now in draft form with the Wheeler group to examine the source of the enantioinduction (submitted for review only). We anticipate that paper helping clarify the issue at hand, and we will use it as a basis for future investigations in quaternary carbon center formation.

--The value of the synthetic application of the method was not shown very well. It is suggested that the authors apply the methodology to the synthesis of natural products or drug molecule analogues (Figure 1) to better demonstrate the value of the methodology.

RESPONSE:

We are indeed making this application, and we will certainly report the successful conclusion of those efforts in a publication dedicated to that project. Both space and time limitations prevent its inclusion in this communication.

Minor revision: Manuscript:

(1) Line 74: "Heteroaromatic enone substituents were likewise accommodated (see 3f)." There is only one example where the singular form should be used.

RESPONSE:

We have made the recommended change.

(2) Compound numbers should be bolded, please check the full text.

RESPONSE:

We have made the recommended change throughout the text.

(3) Line: 119-120: "4" should be bolded, "PhCF₃" should be subscripted.

RESPONSE:

We have made the recommended change.

(4) Line 167: The structure of compound 5a' in figure 4 may be wrong.

RESPONSE:

We appreciate the reviewer pointing out the error, and we have corrected it.

(5) Line 176: 3p should be changed to 3q.

RESPONSE:

We have made the recommended change.

(6) Line 186 (Z)-1,4-enediketones "Z" should be italicized, please check the full text.

RESPONSE:

We have made the recommended change throughout the text.

(7) The format of the references is not standardized according to NC Guideline. Some errors in the format of the references. Ref. 3, 4, 12, 13, 19, 22-33, 35, 36, 37, 49, 50, 62, 66-73, 72, 73, 76.

RESPONSE:

We have made corrections to the listed references.

Supporting Information

(8) Data of the HPLC: retention time should be pointed.

RESPONSE:

The retention times have been updated.

(9) 19F NMR spectra of 1d, 1o, 3d and 3o should be provided.

RESPONSE:

We have made the recommended addition.

(10) The X-ray crystallography of 8 is missing in SI.

RESPONSE:

We have made the recommended addition to an endnote in the manuscript

(11) Specific rotation data of the products are not available.

RESPONSE:

We have made the recommended addition of rotations to the SI.

(12) The yields of 3j, 3l, 3r, 3y in SI are different from the yields in the manuscript. The authors should check the yields of all products.

RESPONSE:

That for 3r was corrected in the manuscript (91%). The other yields have been corrected in the SI.

(13) There are some impurities appeared in the NMR spectra, it is recommended to do further purification: 1b, 3n, 3aa.

RESPONSE:

We have obtained spectra without the impurities noted.

REVIEWERS' COMMENTS

Reviewer #1 (Remarks to the Author):

The authors have addressed my concerns and revised the manuscript properly. For these reasons, I now recommend publication in Nature Communications.

Reviewer #2 (Remarks to the Author):

I am satisfied with the improvements.

Reviewer #3 (Remarks to the Author):

I double-checked the revised manuscript. Although the authors revealed the reactivity and regioselectivity of (Z)-enediketones, and studied the reaction mechanism, enantioselective control of the reaction is equally important. Because highly enantioselective acyclic all-carbon quaternary stereocenters have never been synthesized by the classical 1,4-addition of boric acid or derivatives. It is obvious that the results presented in the manuscript do not have good substrate applicability. Most examples have enantioselectivities below 85% ee. I don't see any changes in the revised manuscript. In general, the question arises whether the outcome will be of interest to the readership of Nature Communications.

I currently recommend the publication of this manuscript in a more specialized journal. If the authors could either re-optimize the conditions to further improve the stereoselectivities of the products (extend the scope of substrates) or if they could provide examples for important applications, it might be re-considered for publication in Nature Communications.

My last comments.

“However, the enantioselectivities of this work are currently not well resolved. Only five products had $\geq 90\%$ ee. Although the authors tried 31 examples, the enantioselectivities of more than 1/3 of the products (12 examples) were below 80% ee. Some examples (3j, 3m, 3n, 3q, 3s, 3t, 3u, 3w, 3x, 9 examples) require a 40 mol% catalyst. The amount of catalyst is too much, and the authors should

consider this point. The authors should re-optimize the conditions to further improve the stereoselectivities of the products.”

“It is suggested that the authors apply the methodology to the synthesis of natural products or drug molecule analogues (Figure 1) to better demonstrate the value of the methodology.”

Reviewers' Comments:

Reviewer #1 (Remarks to the Author):

COMMENT: The authors have addressed my concerns and revised the manuscript properly. For these reasons, I now recommend publication in Nature Communications.

Reviewer #2 (Remarks to the Author):

COMMENT: I am satisfied with the improvements.

Reviewer #3 (Remarks to the Author):

COMMENT: I double-checked the revised manuscript. Although the authors revealed the reactivity and regioselectivity of (Z)-enediketones, and studied the reaction mechanism, enantioselective control of the reaction is equally important. Because highly enantioselective acyclic all-carbon quaternary stereocenters have never been synthesized by the classical 1,4-addition of boric acid or derivatives. It is obvious that the results presented in the manuscript do not have good substrate applicability. Most examples have enantioselectivities below 85% ee. I don't see any changes in the revised manuscript. In general, the question arises whether the outcome will be of interest to 85% ee. I don't see any changes in the revised manuscript. In general, the question arises whether the outcome will be of interest to the readership of Nature Communications.

I currently recommend the publication of this manuscript in a more specialized journal. If the authors could either re-optimize the conditions to further improve the stereoselectivities of the products (extend the scope of substrates) or if they could provide examples for important applications, it might be re-considered for publication in Nature Communications. My last comments. "However, the enantioselectivities of this work are currently not well resolved. Only five products had $\geq 90\%$ ee. Although the authors tried 31 examples, the enantioselectivities of more than 1/3 of the products (12 examples) were below 80% ee. Some examples (3j, 3m, 3n, 3q, 3s, 3t, 3u, 3w, 3x, 9 examples) require a 40 mol% catalyst. The amount of catalyst is too much, and the authors should consider this point. The authors should re-optimize the conditions to further improve the stereoselectivities of the products."

"It is suggested that the authors apply the methodology to the synthesis of natural products or drug molecule analogues (Figure 1) to better demonstrate the value of the methodology."

RESPONSE: We agree with the reviewer that enantiocontrol is important in the development of asymmetric chemical methods, though we are surprised that the reviewer was not able to see the changes made from the first submission. We believe that the reviewer's main point for the need to improve the overall selectivity is valid, and addressing it is the main thrust of our ongoing efforts. The key to this effort will be suppressing the background reaction relative to the catalyzed reaction, since it appears that the ouroboros stabilization is so effective at activating this system that the quaternary carbons are formed even without catalyst! We are consequently exploring many approaches such as dual catalysis, controlled reagent addition, and microwave irradiation to diminish this background reaction, and the discovery of the active transition state will aid in a rational approach. We will report these efforts and improved er values in the full article focused on this effort. We note that we could have met the reviewer's quota of higher er results by producing a dozen examples of the motif that was most successful as is done in many other catalysis papers. Neither we nor the reviewer (most likely) would find that approach useful or of scientific value. We note that this has been the focus of a discussion of the greater synthetic chemistry community (Kozlowski, M. C. On the Topic of Substrate Scope. *Org. Lett.* **2022**, *24* (40), 7247–7249. <https://doi.org/10.1021/acs.orglett.2c03246>), where the leaders of the field agree with us that a broad scope showing a variety of structures, including those that do not provide ideal data, is of the most value to science, rather than a focus purely on appealing numerical data. Thus, we acknowledge where there is room for improvement by the

scientific community, and we are currently vigorously pursuing this.

The compelling point of this report, though, is the remarkable catalytic activation of this substrate that may be exploited in a variety of reactions. The knowledge of these stabilizing effects are applicable to self-assembled materials, enzymatic catalysis, autocatalytic reactions, molecular photophysics, and any other application that relies on molecular structure and energy. Informally, colleagues who have seen these results have been ecstatic and highly intrigued, and we have been urged to publish them ASAP. We believe strongly that this disclosure will drive innovation and applies to fields as diverse as biochemistry, protein biology, medicinal chemistry, material science, and catalysis. Because of this, the work merits rapid publication to enable its access to those practitioners while we finalize the optimization of reaction parameters. Even now, though, most all of the examples are synthetically useful and demonstrate the wide array of molecular motifs that are compatible, which is a significant improvement over transition metal catalysis.